# Assessing Knowledge, Uptake and Factors associated with cervical cancer screening among women in selected communities of Wakiso district in Uganda: A population-based study

Robert M. Bulamba[1]*, Emmanuel Kyasanku[1], Fred Nalugoda[1], Alex Daama[1], James Nkale[1], Amanda Pearl Miller[2], William Byansi[3], Juliana Namutundu[4], Godfrey Kigozi[1], Grace Kigozi Nalwoga[1], Chris Balwanaki[4], Stephen Watya[1], Anna Mia Ekström[5], Stephen Mugamba[1], Rawlance Ndejjo[4], Gertrude Nakigozi[1]

1 Department of Epidemiology and Biostatistics, Africa Medical and Behavioral Sciences Organization (AMBSO), Kampala, Uganda, 2 Department of Epidemiology and Biostatistics, San Diego State University, San Diego, California, United States of America, 3 Department of Social Work, School of Social work Boston College, Chestnut Hill, Massachusetts, United States of America, 4 Department of Disease Control and Environmental Health, Makerere University School of Public Health (MakSPH)), Kampala, Uganda, 5 Department of Infectious Diseases, Karolinska Institute, Stockholm, Sweden

* rbulamba@ambso.org

## Abstract

### Background

Uganda has the highest prevalence and incidence of cervical cancer in the East African region, with 80% of women diagnosed at advanced stage when survival is minimal. Literature on uptake of cervical cancer screening is limited in Uganda and thus womens' knowledge and uptake of cervical cancer screening in the general population remains unknown. This study examined this gap of knowledge among women aged 25–65 years, across rural, urban and semi urban communities in a Ugandan district to inform design of targeted future cervical cancer screening programs in the country.

### Methods

This descriptive cross-sectional study was conducted in Wakiso district, Uganda in May 2024 among 783 eligible women. Face-to-face interviews were conducted. Uptake of cervical cancer screening (outcome of interest) was dichotomously (yes/no) assessed. Knowledge of cervical cancer disease was assessed using the AWA-CAN validated tool, knowledge of cervical cancer screening was assessed using a set of ten (10) questions adapted from previous studies elsewhere, and all were measured on a Likert scale. Univariate, bivariate, and multivariable Poisson regression models with robust variance were performed using Stata software version 17.

**Data availability statement:** All relevant data are within the paper and its Supporting Information files.

**Funding:** This work was supported by the National Institute of Health and Care Research (NIHR), through the Royal Society of Tropical Medicine and Hygiene (RSTMH) Early Careers Research Program, grant number: NIHR170.

**Competing interests:** Authors declare no competing interests.

## Results

Respondents' median age was 31 years (IQR 27–39 years). Majority (89.5%, 701/783) had heard of cervical cancer, and 90.6% (635/701) were aware of screening. Median knowledge score on signs and symptoms, risk factors and cervical cancer screening was 8.0 (IQR = 5–10), 8.0 (IQR = 5–11) and 7.0 (IQR = 4–10) respectively, and 54.3% had high knowledge about cervical cancer screening. Uptake of cervical cancer screening was 33.4%. Living in urban areas (aPR = 1.41, 95% CI: 1.05–1.88), being the ages 40–49 years (aPR = 1.76, 95% CI: 1.36–2.27), 50 years and above (APR = 2.16, 95% CI: 1.53–3.04), smoking (aPR = 1.39, 95% CI: 1.05–1.86), partner involvement (aPR = 2.61, 95% CI: 2.12–3.21), high knowledge about cervical cancer screening (aPR = 3.29, 95% CI: 2.35–4.60), and living with HIV (aPR = 1.66, 95% CI: 1.66–2.13) were significantly associated with higher uptake of cervical cancer screening among women in this setting.

## Conclusion

Knowledge of cervical cancer screening was high, but the uptake of cervical cancer screening was lower than the recommended population coverage by WHO and Uganda national guidelines. There is need to improve accessibility to cervical cancer screening, increase nationwide cervical cancer awareness campaigns focusing on high-risk age groups and design targeted, tailored, culturally and socially sensitive interventions for young women aged 25–39 years to improve cervical cancer screening in Uganda.

## Introduction

Cervical cancer is cancer of the cervix, the lower part of the uterus that connects to the vagina (CDC report, 2023). Cervical cancer develops from the cells of the cervix, the narrow passage connecting the uterus to the vagina. it typically progresses over many years, often starting with precancerous changes that can be detected through screening [1,2]. Cervical cancer is primarily caused by persistent infection with high-risk strains of the human papillomavirus (HPV). Additional risk factors of cervical cancer include: smoking, which weaken the immune system, multigravida, early onset of sexual activity, having multiple sexual partners, and a family history of cervical cancer, among others [3,4]. Globally, remarkable progress has been achieved in the prevention and control of Cervical Cancer [5]. However, although preventable and curable, it is still the fourth most common cancer among women globally and the leading cause of gynecologic cancer-related deaths among women in low and middle income countries (LMICs) [1], yet uptake of screening services is still low ~ 36% in LMICs below the 70% global coverage target [6].

In sub-Saharan Africa (SSA), ~ 78, 897 women are diagnosed with cervical cancer, with prevalence in the region varying from 10% – 40% annually. These cases lead to 61,539 (78%) cervical cancer-related deaths annually and young woman are more

affected than in any other region of the world [7]. Despite this burden, the uptake of cervical cancer screening services in SSA is still low at ~ 25% [8]. In Uganda, cervical cancer remains the number one cause of cancer-related death in women and it is responsible for ~ 6,959 cases and 4,607 deaths annually [9]. The incidence is three [3] times that of the global average and mortality is exceptionally high with 41.4 deaths per 100,000 women age-standardized mortality, and a 17.7% 5-year relative survival for cervical cancer [10]. Uganda's Ministry of Health has set a national screening target of 70%, but uptake of cervical cancer screening services in the country remains way below this target at ~10% [11]. Despite the availability of modern treatment options and prevention strategies (e.g., access to the HPV vaccine series) in Uganda, cervical cancer screening uptake is still low with lifetime screening estimates ranging between 4.8% − 35.1% among women aged 25–49 years in the country [12,13].

Cervical cancer screening is one of the effective preventive measures to combat cervical cancer disease among the women population [14]. Early screening allows for early-stage detection of cervical cancer diagnosis, which facilitates prompt non-aggressive treatment that promotes health, prolongs life and increases survival [15,16]. Cervical cancer is highly preventable, but poor access to prevention, screening and treatment contributes to 90% of the deaths, and majority of the cervical cancer-related deaths occur due to late screening when survival is minimal [17], due to lack of knowledge about cervical cancer and screening as well as limited availability of the screening programs in Uganda. Additionally, there exists limited data regarding cervical cancer screening at population level in Uganda, particularly in Wakiso district, which limits the design of high-impact community-based cervical cancer screening interventions to reduce the incidence of cervical cancer in the district. The Uganda MOH recommends cervical cancer screening for women aged 25–49 years, however, the current WHO guidelines highlight a gap in data regarding cervical cancer screening among women aged 50–65 years, hence the inclusion of this population strengthens the relevance of our study by offering insightful literature in the Ugandan context.

## Materials and methods

For the present cross sectional study, we nested questions into the fifth round of the Africa Medical and Behavioral Sciences Organization (AMBSO) Population Health Surveillance (APHS) study [18]. Briefly, the APHS is a longitudinal community-based cohort study, being implemented across six communities in Wakiso and Hoima districts in Uganda. APHS study communities were chosen from a larger sample following a community mapping exercise comparing the population structure of multiple districts with different distances from urban centers and varying degrees of cultivation for their representativeness of three different community types (urban, semi-urban and rural) to improve the generalizability of the cohort's data to Uganda's broader population [18].

For the present study, we integrated additional questions around cervical cancer knowledge and uptake to the APHS survey and administered it to a systematically selected subset of participants in the three Wakiso district APHS communities, leveraging the cohort studies' census activities and access to a large generalizable sample. These survey communities, Sentema village (rural), Kazo cell (urban) and Lukwanga town (semi-urban) are all approximately 27 kilometers from Kampala, Uganda's capital, and largest city. Data for the present study was collected from 06th – 15th April 2024. APHS participants who were women aged 25–65 years and residents of the study community for at least six (6) months at the time of the study were eligible. Our study target population was guided by the present Uganda national guidelines which target screening women starting from the age of 25 years.

The APHS census database was used to establish the total number of eligible participants in each of the study communities. According to the APHS census cohort data, Sentema the rural community had a total population of 2,094 women of whom, 627 (30%) were eligible women aged 25–65 years. Lukwanga the semi-urban community had a total population of 4,816 women, of whom 1,372 (28%) were eligible women aged 25–65 years of age. Kazo the urban community had a total population of 2,922 women, of whom 862 (30%) were eligible women aged 25–65 years.

## Sample size

The study sample size was determined using the Kish Leslie's formula (1965).

$$\text{Sample Size (n)} = \frac{P\,(1-P)\,Z_2}{d^2}$$

$$\text{Sample Size (n)} = \frac{0.5^*\,(1-0.5)^*\,(1.96)^2}{(0.05)^2}$$

Where P = estimated proportion of cervical cancer screening among women (50%), as this provided the maximum possible sample size for more precise estimates of the study findings, Z = statistic corresponding to 95% Confidence level = 1.96 and d = margin of error (5%). Therefore, the sample, n = 384 participants. To adjust for sampling design and improve the precision of our estimates, we calculated the design effect as follows; Design effect (Deff) = 1 + δ(n-1), where, δ = intercluster correlation (ICC) and n = number of clusters. We used δ = 0.5 recommended in previous work by [19] as a moderately good ICC for estimating representative sample sizes in one time surveys. Deff = 1 + 0.5(3−1) = 2. Therefore, the Sample Size = 384* Deff (2) = 768. Estimated non response rate was 5%, making the total study sample size of 806 women. Overall, our calculated sample was 806 inclusive of the 5% (38) non-response rate. Although our target sample size was 768 participants, this sample was exceeded by 15 participants during study enrolment, hence obtaining a sample of 783 participants. For this analysis, all the 783 enrolled participants were included. We then applied Probability Proportional to Size (PPS) sampling approach to obtain the required number of study participants from each of the three (3) selected communities, as shown in table below.

| Category | Communities | | | |
|---|---|---|---|---|
| **Calculated sample size = 806** | **Kazo (urban)** | **Sentema (rural)** | **Lukwanga (S.urban)** | **Total** |
| Total target population | 862 | 627 | 1,372 | 2,861 |
| Sample size calculation | $\frac{862*806}{2,861}$ | $\frac{627*806}{2,861}$ | $\frac{1,372*806}{2861}$ | |
| **Total sample size** | **243** | **177** | **386** | **806** |
| **No. of households reached** | $\frac{862=216}{4}$ | $\frac{627=157}{4}$ | $\frac{1372=343}{4}$ | **716** |

The study assumed an average household (sampling unit) size of 4 in each of the study communities. Approximately, the total number of households in each of the study communities from which the eligible participants were drawn was estimated as follows; 216 participants in Kazo, 157 participants in Sentema and 343 participants in Lukwanga. Systematic random sampling was applied to select sampling units in the study communities from which participants were drawn(20). The total required sample was then divided by the total target population for each of the study communities to obtain the sampling interval as follows; Kazo = 862/243 = 4, Sentema = 627/177 = 4 and Lukwanga = 1,372/386 = 4 households. Therefore, every 4th household in each of the study communities was selected. At the sampling units (households), every participant who was eligible and willing to participate was interviewed to maximize chances of obtaining the required number of women aged 25–65 years in each study community. Data collection used a paper-based structured questionnaire with closed ended questions adapted from tools used in studies elsewhere [12,19,20]. We collected data on socio-demographic characteristics of women including age, religion, educational status, marital status, community type (area of residence) and type of employment, reproductive factors such as sexual activity, age at first sexual debut, age at first marriage, number of children (parity) and others, data on sexual-behavioral and family planning practices such as contraceptive use, number of sexual partners, HIV testing and HIV test results, and others, data on cervical cancer disease and HPV, data on cervical cancer screening, as well as data on partner involvement. This study was approved by

the Makerere University School of Public Health – Research and Ethics Committee (MakSPH-REC). Participation in the research study was voluntary, and written informed consent was obtained from all participants prior to their participation.

Data was collected by 15 research assistants (10-females and 5-males). The research assistants (RAs) were trained for 5 days (1st - 5th April 2024) on principles of quantitative and survey research including data collection, consenting process, the research study objectives, and procedures for; sampling and assessing eligibility for study participation, and the research data tools were translated in Luganda-the local language. For validation purposes and checking for adaptability and understanding of the questions, consistence and flow of the questions, the data collection tools (questionnaires) were pre-tested on 10 women participants who were not living within the communities where the study was conducted. The 10 participants were as well reimbursed for their time during the pilot testing of the tool. After the pre-testing any required adjustments were made and then the research team was re-trained on the adjustments in the tools.

## Measures

Knowledge of cervical cancer was measured using questions adapted from the modified African Women Awareness of CANcer (AWACAN) instrument [19]. The AWACAN tool has been described elsewhere [21]. Briefly, the tool is a 41-item questionnaire that measures women's knowledge of cervical cancer. The tool measures cancer related knowledge in the following domains: risk factors, signs and symptoms, lay beliefs, confidence in appraisal, health-seeking behaviours, barriers to health care and socio-demographic characteristics. The tool was validated with two modules on risk factors, and signs and symptoms of cervical cancer [19], particularly for this study these two modules were adapted for use. Responses for the risk factors (15 items) and signs and symptoms (12 items) domains were used in the present analysis. Both modules were assessed among respondents who reported having ever heard about cervical cancer as per the AWACAN tool. The responses in each of the 2 modules were: Yes/No/Don't know. For each correct response on an item/question in the module, a score of 1 was assigned and a score of 0 was assigned to a wrong response [19]. Therefore, for each of the 2 modules, the scores were summed up and composite variables for risk of cervical cancer, as well as signs and symptoms were generated. A binary variable was created from each of these continuous measures using the median to divide participant responders into two categories, those with "low knowledge" scores below the median score and "high knowledge" with scores>= to the median scores. The summed up continuous variables were used in the descriptive analysis, while the binary variables was used in the bivariate and multivariable regression models.

Knowledge of cervical cancer screening was measured by a set of ten (10) questions adapted from previous studies [19,22,23]. These questions examined women's specific knowledge about cervical cancer disease and cervical cancer screening. One point was given if a respondent gave one or more correct response(s) on each item/question in the module [19]. As with the cervical cancer risk factors and signs and symptoms variables described above, we obtained a composite knowledge variable which was then converted to a binary variable using the median as the cutoff [22,23]. Women who obtained scores greater than the median were considered to have high knowledge and those with scores less than the median score were considered to have low knowledge [19,22,23].

## Data preparation and statistical analysis

Data collected on paper questionnaires was keyed into a designed Open Data Kit (ODK) application system in ODK Central v2023.5.1 hosted at the AMBSO server at its research complex facility in Nansana, Wakiso district- Uganda. Data was then downloaded to excel as a csv file, cleaned, and then exported to Stata version 17.0 [24] for analysis. Using the Shapiro Wilk test, normality was assessed for the continuous variables such as; age, knowledge of cervical cancer signs and symptoms, risk factors and cervical cancer screening, and thus non-parametric test results are reported accordingly. The test reliability and internal consistency for knowledge of cervical cancer risk factors was 0.89, and suggested a strong internal consistency of the measure's ability to effectively assess women's knowledge about cervical cancer risk factors, consistent with 0.60 obtained in prior work [19]. Similarly, the reliability and internal consistency of the tool's measure

for knowledge of signs and symptoms was assessed, and a Cronbach's alpha of 0.86 was obtained, consistent with 0.80 obtained in a previous validation study [19]. This suggests a strong reliability and internal consistency of the tool to effectively measure women's knowledge about cervical cancer signs and symptoms as well as risk factors. Multicollinearity between predictors such as overall knowledge for cervical cancer and knowledge for cervical cancer screening (was 2.07) and between knowledge of cervical cancer signs and symptoms with risk factors (was 1.14) was checked using the variation inflation factor (VIF), which suggested absence of multicollinearity between these predictors in line with previous models [25]. Since the prevalence of cervical cancer screening among women was 33.4%, we fitted a modified Poisson multivariable regression model with robust standard errors to determine the factors associated with uptake of cervical cancer screening among women. After the bivariate analysis, all factors with a P-value <=0.05 were included in the final regression model as potential factors associated with the outcome, adjusting for to limit bias from confounding. In the presentation of results, unadjusted and adjusted Prevalence Ratios (uPRs) and (aPRs) were interpreted and reported with accompanying 95% confidence intervals.

## Results

### Sociodemographic characteristics of the study sample

Majority (70.4%, 551/783) of women were married at the time of the survey, and more than half (76%, 594/783) were between the ages of 25–39 years, with a median age of 31 (IQR 27–39). The majority of women (53.4%, 418/783) had attained at least a secondary level of education, and a sizeable minority of the women (47.5%, 372/783) resided in the semi-urban community (Table 1).

Overall knowledge about cervical cancer disease and cervical cancer screening was high. Majority (54%, 376/701) of participants had high knowledge about cervical cancer signs and symptoms, median score = 8 (IQR = 5–10). More than half (55.2%, 387/701) of the participants had high knowledge about risk factors of cervical cancer, with median = 8 (IQR = 5–11). Overall, majority (54.8%, 384/701) of the women had high knowledge about cervical cancer disease, with median score = 15 years and IQR = 11–20 years. Women's knowledge about cervical cancer screening was high (53.4%, 374/701), with median score 7 years and IQR = 4–10 years (Table 2).

Awareness about cervical cancer screening was high (90.6%) among women in this study, ~86% of the women believed that when a women is diagnosed early, they stand a chance of healing from the disease, and 97.2% of them believed that a cervical cancer screening test is supposed to be performed by a trained health worker. Almost half (49.6%) of the women believed that women aged below 25 years are highly recommended for cervical cancer screening. 86.5% of women at least knew of a place/facility where to go for cervical cancer screening, more than half (62.7%) reported that their homes are <=5 kilometers to the cervical cancer screening facilities. Majority (60.3%) of the women reported that they obtained initial information about cervical cancer screening from health care workers, and 10.2%, 6.9%, 5% and 9.6% obtained information from radios, family and friends, teachers/religious leaders and then televisions respectively (Table 3).

Fig 1 shows the relationship between knowledge about cervical cancer and knowledge about cervical cancer screening. A few participants had knowledge about cervical cancer disease, and as observed, the Knowledge about cervical cancer screening increased with increasing knowledge about cervical cancer disease in this population.

**Uptake of cervical cancer screening among women aged 25–65 years in Wakiso district, Uganda.** Among respondents who reported ever screening for cervical cancer (33.4%) in the past 5 years, majority (20.16%) were aged 25–39 years, 8.35% were aged 40–49 years and 4.88% were aged 50 years and above. Among respondents who reported that they have never screened for cervical cancer (66.6%), more than half of them (55.28%) were aged 25–39 years, followed by those aged 40–49 years (6.61%) and those aged 50 years and above with 4.72% (Fig 2).

**Factors associated with cervical cancer screening among women in Wakiso district.** Results from the bivariate analysis (Table 4) indicated that age, residence, history of smoking, number of sexual partners, age at first marriage,

**Table 1. Sociodemographic characteristics of the study sample.**

| n = 783 | |
|---|---|
| **Variable** | **Frequency, n (%)** |
| **Age group**[Median = 31, (IQR = 27–39)] | |
| 25–39 years | 594 (76.0) |
| 40–49 years | 114 (14.5) |
| 50 years + | 75 (9.5) |
| **Education Level** | |
| None/ Primary education | 365 (46.6) |
| Secondary and above education | 418 (53.4) |
| **Religion** | |
| Catholics | 275 (35.1) |
| Protestants | 207 (26.4) |
| Moslems | 162 (20.7) |
| Saved/Pentecostal | 126 (16.1) |
| Others (Seventh day Adventists (SDA)) | 13 (1.7) |
| **Marital Status** | |
| Married | 551 (70.4) |
| Single | 232 (29.6) |
| **Community Type** | |
| Urban | 236 (30.0) |
| Rural | 175 (22.4) |
| Semi-urban | 372 (47.5) |
| **Employment Type** | |
| Agriculturists | 196 (25.0) |
| Housework in own home | 423 (54.0) |
| Trading/ Vending | 79 (10.0) |
| Waitress/ Saloon | 38 (5.0) |
| Other | 48 (6.0) |

*Other Religion* includes: SDA and none *Other employment type* includes: gov't and medical workers, truck drivers, oil workers. *Single* also includes those who divorced or widowed

number of children, HIV status, seeking permission from partners to go for cervical cancer screening and knowledge about cervical cancer screening were significantly associated with cervical cancer screening uptake among women in this study. The prevalence of cervical cancer screening was 2.10 times high among women aged 40–49 years compared to those aged 25–39 years of age (uPR = 2.10, 95% CI:1.76–2.63). Similarly, the prevalence of cervical cancer screening was 1.90 times higher among older women (50 years and above) compared to women aged 25–39 years (uPR = 1.90, 95% CI:1.42–2.54).

The prevalence of cervical cancer screening among women living in the urban areas was 66% higher compared to that of women living in the rural areas (uPR = 1.66, 95% CI:1.19–2.29). Women with a history of smoking had a prevalence that was 74% higher than that of women without a history of smoking in this study (uPR = 1.74, 95% CI:1.33–2.26). Women who reported using family planning, those with one sexual partner (past 12 months) and those who got married while they were aged 18–24 years had prevalence of 0.73, 0.64 and 0.74 times lower compared to those who were not using family planning, those without a sexual partner and those who got married while they were 13–17 years respectively. Furthermore, the prevalence of cervical cancer screening was 2.39 times higher among women with three (3) children and more

**Table 2. Knowledge about cervical cancer and screening among women in Wakiso district, Uganda.**

| Variable | Frequency, (%) |
|---|---|
| Knowledge about cervical cancer Signs and Symptoms Likert Scale Median score = 8 [IQR = 5–10] | |
| Low Knowledge [0–7] | 322 (46) |
| **High Knowledge [8 and above]** | **379 (54)** |
| Knowledge about cervical cancer Risk Factors Likert Scale Median score = 8 [IQR = 5–11] | |
| Low Knowledge [0–7] | 314 (44.8) |
| **High Knowledge [8 and above]** | **387 (55.2)** |
| Overall Knowledge about cervical cancer disease Likert Scale Median Score = 15, [IQR = 11–20] | |
| Low Knowledge [0–14] | 317 (45.2) |
| **High knowledge [15 and above]** | **384 (54.8)** |
| Knowledge about cervical cancer Screening services among women Likert Scale Median score = 7 [IQR = 4–10] | |
| Low Knowledge [0–6] | 327 (46.7) |
| **High Knowledge [7 and above]** | **374 (53.4)** |

compared to those with no child (uPR = 2.39, 95% CI:1.13–5.02). The prevalence of cervical cancer screening was 1.57 times higher among women with high knowledge about cervical cancer screening compared to that of women with low knowledge (uPR = 1.57, 95% CI:1.20–1.94), and it was 2.62 times higher among women living with HIV compared to those without HIV (uPR = 2.62, 95% CI:2.15–3.19).

Results indicated that the prevalence of cervical cancer screening among women aged 40–49 years was 1.76 times higher compared to that of women aged 25–39 years (aPR = 1.76, 95% CI: 1.36–2.27). Additionally, the prevalence of cervical cancer screening among older women aged 50 years and above was 2.16 higher compared to that of women aged 25–39 years (aPR = 2.16, 95% CI: 1.53–3.04).

Results showed that women living in urban areas or communities had a cervical cancer screening prevalence of 1.41 higher compared to that of women living in the rural areas (aPR = 1.41, 95% CI: 1.05–1.88). There were no significant difference in cervical cancer screening uptake among women living in the semi urban areas when compared to those living in rural areas (aPR = 1.04, 95% CI: 0.79–1.36). Furthermore, the prevalence of cervical cancer screening was 1.39 higher among women with a history of tobacco, cigarettes, or pipe smoking compared to women without a history of smoking (aPR = 1.39, 95% CI: 1.05–1.86).

Women with one sexual partner in the past 12 months had a 0.63 lower prevalence of cervical cancer screening compared to those with no sexual partners (aPR = 0.63, 95% CI: 0.40–0.99). Furthermore, the prevalence of cervical cancer screening among women who sought permission from their partners about cervical cancer screening was 2.61 times higher compared to that of women did not seek permission from their partners (aPR = 2.61, 95% CI: 2.12–3.21). Additionally, the prevalence of cervical cancer screening was found to be to 3.29 times higher among women with high level of knowledge about cervical cancer and cervical cancer screening compared to those with low levels of knowledge (aPR = 3.29, 95% CI: 2.35–4.60). We also observed that the prevalence of cervical cancer screening was 1.66 times higher among women living with HIV compared to that of HIV-negative women (aPR = 1.66, 95% CI: 1.29–2.13) in this study setting (Table 4).

## Discussion

This study provides information on the level of knowledge, uptake, and factors associated with cervical cancer screening among women aged 25–65 years residing in diverse communities in Uganda. In this study, knowledge about signs and

**Table 3. Knowledge questions about cervical cancer screening.**

| Variable | N=701 | |
|---|---|---|
| | **Frequency** | **Percent %)** |
| **Knowledge about cervical cancer screening among women** | | |
| Have you ever heard about cervical cancer screening? | | |
| Yes | 635 | 90.6 |
| No | 66 | 9.4 |
| **Selected specific Knowledge questions about cervical cancer screening** | | |
| Circumstances under which a woman diagnosed with cervical cancer stand high chances of healing? | | |
| When diagnosed late | 4 | 1.8 |
| When diagnosed early | 191 | 86 |
| At any time of diagnosis | 17 | 7.7 |
| I don't know | 9 | 4 |
| In your view who is supposed to perform a cervical cancer screening test? | | |
| Any Person | 14 | 2.2 |
| A trained health worker | 617 | 97.2 |
| Traditional healer | 4 | 0.7 |
| Rate the importance of a woman attending cervical cancer screening in her life time? | | |
| Very important | 421 | 66.3 |
| Important | 214 | 33.7 |
| Age category of women are recommended for cervical cancer screening test? | | |
| Any age | 156 | 24.6 |
| Below 25 years | 315 | 49.6 |
| 25–49 years | 138 | 21.7 |
| Above 49 years | 15 | 2.4 |
| Don't know | 11 | 1.7 |
| Know of any place where people can go for cervical cancer screening? | | |
| Yes | 549 | 86.5 |
| No | 86 | 13.5 |
| Walking distance in Kms from home to the cervical cancer screening facility? | | |
| <=5 KMs | 344 | 62.7 |
| 6–10 KMs | 75 | 13.7 |
| >=11KMs | 130 | 23.7 |
| **Source of information about cervical cancer Screening** | | |
| How did you first learn about cervical cancer screening? | | |
| Radio | 65 | 10.2 |
| Family and Friends | 44 | 6.9 |
| Healthcare workers | 384 | 60.3 |
| Teachers/Religious leaders | 32 | 5.0 |
| Television | 61 | 9.6 |
| Other **(posters, billboards, brochures)** | 50 | 7.9 |
| **Awareness about Cervical cancer** | | |
| Yes | 701 | 89.5 |
| No | 82 | 10.5 |

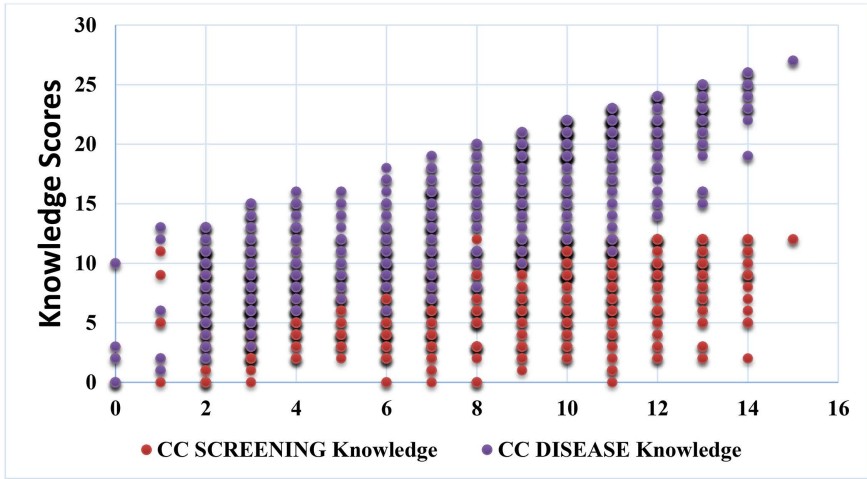

**Fig 1. Knowledge about cervical cancer (CC) and Screening among women in Wakiso district.**

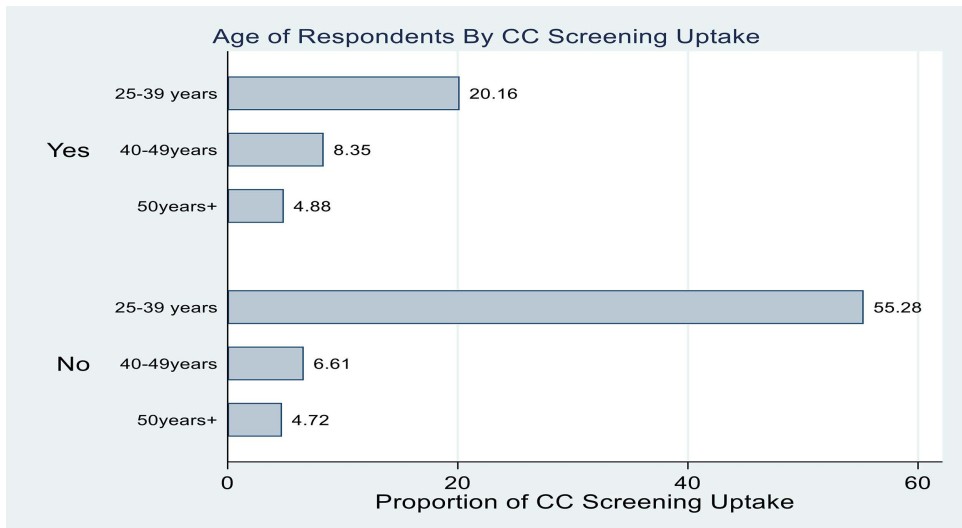

**Fig 2. Showing women's cervical cancer screening rates per age group.**

symptoms as well as risk factors for cervical cancer disease was high. Similarly, participants had high knowledge about cervical cancer screening and knowledge about cervical cancer screening increased with increase in knowledge about cervical cancer disease. Women living with HIV, those with higher knowledge about cervical cancer screening, those who involved their partners while making decisions about cervical cancer screening were associated with higher prevalence of cervical cancer screening in this study.

The high levels of cervical cancer knowledge observed in our sample is consistent with extant research in the region (northern Uganda [26] and Kenya) [25]. In contrast, a recent [27] study in north central Nigeria involving federal civil servants found much lower rates of both cervical cancer awareness (38.6%) and screening (10.2%) with most of those who had received screening, doing so as part of routine antenatal testing. This could have been due to the fact that the major

**Table 4. Association between cervical cancer screening, behavioral, reproductive, and other covariates.**

Modified Poisson Regression Model with Robust variance

| Factor Variable | Uptake of Cervical Cancer Screening | | | | | |
| --- | --- | --- | --- | --- | --- | --- |
| | Unadjusted Prevalence Ratio(uPR) | 95% Confidence Interval (CI) | P-Value | Adjusted Prevalence Ratio(aPRs) | 95% Confidence Interval (CI) | P-Value |
| **Age group** | | | | | | |
| 25–39 years | (ref) | (ref) | (ref) | (ref) | (ref) | (ref) |
| 40–49 years | 2.10 | 1.76-2.63* | <0.001 | 1.76 | 1.36-2.27* | <0.001 |
| 50 years and above | 1.90 | 1.42-2.54* | <0.001 | 2.16 | 1.53-3.04* | <0.001 |
| **Residence** | | | | | | |
| Rural | (ref) | (ref) | (ref) | (ref) | (ref) | (ref) |
| Urban | 1.66 | 1.19-2.29* | 0.002 | 1.41 | 1.05-1.88* | 0.021 |
| Semi-urban | 1.14 | 0.82-1.59 | 0.443 | 1.04 | 0.79-1.36 | 0.782 |
| **Ever smoked Tobacco, cigarette or pipe** | | | | | | |
| No | (ref) | (ref) | (ref) | (ref) | (ref) | (ref) |
| Yes | 1.74 | 1.33-2.26* | <0.001 | 1.39 | 1.05-1.86* | 0.024 |
| **Currently using family planning** | | | | | | |
| No | (ref) | (ref) | (ref) | (ref) | (ref) | (ref) |
| Yes | 0.73 | 0.57-0.92* | 0.008 | 0.92 | 0.75-1.16 | 0.514 |
| **Number of sexual partners (Past 12 months)** | | | | | | |
| None | (ref) | (ref) | (ref) | (ref) | (ref) | (ref) |
| One Partner | 0.64 | 0.469-0.88* | 0.006 | 0.63 | 0.40-0.99* | 0.043 |
| Two Partners + [Multiple] | 0.72 | 0.48-1.07 | 0.100 | 0.76 | 0.47-1.24 | 0.27 |
| **Age at first marriage** | | | | | | |
| 13–17 years | (ref) | (ref) | (ref) | (ref) | (ref) | (ref) |
| 18–24 years | 0.74 | 0.58-0.95* | 0.019 | 0.99 | 0.79-1.25 | 0.995 |
| 25 years + | 0.73 | 0.52-1.01 | 0.061 | 0.99 | 0.70-1.39 | 0.957 |
| **Number of Children** | | | | | | |
| No Child | (ref) | (ref) | (ref) | (ref) | (ref) | (ref) |
| 1–2 children | 1.69 | 0.79-3.63 | 0.177 | 1.31 | 0.70-2.43 | 0.396 |
| 3 children and above | 2.39 | 1.13-5.02* | 0.022 | 1.26 | 0.67-2.37 | 0.465 |
| **Relative/ Friend with cervical cancer** | | | | | | |
| No | (ref) | (ref) | (ref) | (ref) | (ref) | (ref) |
| Yes | 1.53 | 1.21-1.93* | <0.001 | 0.85 | 0.67-1.08 | 0.156 |
| **Decision on health** | | | | | | |
| Respondent | (ref) | (ref) | (ref) | (ref) | (ref) | (ref) |
| Partner | 0.13 | 0.58-1.33 | 0.586 | – | – | – |
| Other | 1.53 | 0.92-2.54 | 0.098 | – | – | – |
| **Sought permission from partner to go for cervical cancer screening** | | | | | | |
| No | (ref) | (ref) | (ref) | (ref) | (ref) | (ref) |
| Yes | 3.55 | 2.99-4.21* | <0.001 | 2.61 | 2.12-3.21* | <0.001 |
| **Level of Knowledge about cervical cancer Screening** | | | | | | |
| Low Knowledge | (ref) | (ref) | (ref) | (ref) | (ref) | (ref) |
| High Knowledge | 1.57 | 1.20- 1.94* | <0.001 | 3.29 | 2.35-4.60* | <0.001 |
| **Women's HIV status** | | | | | | |
| Negative | (ref) | (ref) | (ref) | (ref) | (ref) | (ref) |
| Positive | 2.62 | 2.15-3.1* | <0.001 | 1.66 | 1.29-2.13* | <0.001 |

source of cervical cancer information was the media, and therefore, information may have not been literally delivered to influence uptake, which is contrary to Uganda where much of the cervical cancer information is obtained from health care workers who are trained and equipped with right information with proper delivery models. The findings in this study were also not comparable with findings from a study in Nigeria [3] which found absence of any direct relationship between knowledge and uptake of cervical cancer screening services. In addition to this study's findings, cervical cancer screening uptake was higher (49.7% and 55%) among women with high knowledge about cervical cancer signs and symptoms, and risk factors respectively.

One third of the women in our study had undergone cervical cancer screening in the past 5 years, and this is still short of the national targets. Findings from this study indicate a higher uptake of cervical cancer screening compared to the findings from studies in rural districts and fishing communities of Uganda where the uptake was 4.8% and 23.2% [12,28] respectively. The low uptake of cervical cancer screening in these studies could be attributed to the limited availability of healthcare facilities providing cervical cancer screening services in these settings which limits accessibility, and were conducted among women aged only 25–49 years and 15–49 years in which majority do not screen for cervical cancer. The findings from this study are comparable with a study in Uganda [29] among women aged 15–49 years in Oyam district which revealed that 35.1% of the women had ever screened for cervical cancer. The findings are also comparable with findings from a recent study conducted in Kenya by [25] which found an uptake of 28.3%. These results are comparable with findings from a study on AGEing and adult health (SAGE) wave 2 conducted between 2014 and 2015 in Ghana [30] which found that the uptake of cervical cancer screening services was 26.9%. These findings indicate that cervical cancer screening is not practiced by women to the expectations and many women present with late-stage disease when survival is minimal. Although, the findings from this study indicate slightly higher screening rates compared to similar community-based studies conducted previously in Uganda, the study presents screening rates that are far lower than the average coverage of cervical cancer screening (63%) in developed countries [31]. This might be due to the difference in wealth status, educational status, and exposure to the media. Cultural variations might also contribute to the observed discrepancy, as certain cultural norms impose restrictions on women's access to health facilities based on cultural standards. These norms may cause women to perceive cervical cancer screening as a taboo topic, which may reduce the use of screening programs [32].

Although findings on the uptake of cervical cancer screening from our study were lower than the average coverage of cervical cancer screening (33.4% vs 63%) in developed countries, this finding was slightly higher compared to that obtained from other several similar community-based studies in Uganda; A study in Entebbe municipality, Uganda among women aged 25–49 years found a screening rate of 17.1% [33], another study conducted among 900 women in rural districts of Uganda (Bugiri and Mayuge) found a cervical cancer screening uptake of 4.8% [12] and another study in Uganda among 847 women aged 25–49 years found that Only 1 in 5 women (20.6%) had ever screened for cervical cancer [20]. A study among 293 women in fishing communities in Wakiso district found that cervical cancer screening rate was 23.2% [28]. Another study among 656 women aged 25–49 years found a screening rate of 20.3% in Wakiso district [34]. These observed differences could be attributable to the fact that this study focused on women of a wide age range (25–65 years) and the previous studies provided findings from populations of women aged 25–49 years. Though this finding is slightly higher compared to findings from previous studies in Uganda, the cervical cancer screening rate is still far lower than the WHO global and Uganda MOH target for cervical cancer elimination of having 70% of women screened at least two times by age 45 years [35]. Therefore, targeted interventions including but not limited to behavioral change communication and screening programs, should focus on young women aged 25–39 years to modify their behaviors and improve cervical cancer screening in this high-risk population. Additionally, there is need for modifying the current cervical cancer screening practice and incorporate in a community-based screening approach to facilitate early detection of cervical cancer pre-cancerous conditions among women for early initiation of treatment in this setting.

We found significant association between age and cervical cancer screening uptake. Being over 40 years of age was significantly associated with higher cervical cancer screening uptake compared to younger women (25–39 years), which is consistent with studies in central Uganda [22], Kenya [28] Jamaica [36], South Africa [37] and South Ethiopia [38], which found higher uptake of cervical cancer screening among women aged 40 years and older. This study's findings suggest that there might be a potential awareness or recommendation bias towards older age groups, possibly due to increased healthcare interactions or perceived risk among older women. Additionally, this finding suggests that young women do not perceive themselves as at risk of cervical cancer disease and hence do not go for cervical cancer screening services. However this findings are in contrast with findings in Ghana among women aged 20–65 years living with HIV which did not find any significant association between age and cervical cancer screening uptake [39].

This study also found that the prevalence of cervical cancer screening uptake was higher among women living in the urban areas compared women living in the rural areas. Conversely, women in semi-urban areas did not show a significant difference in uptake of cervical cancer services compared to their rural counterparts. The high rates of cervical cancer screening observed particularly in urban areas may have been influenced by availability and quick access to health care services in the urban areas compared to rural/semi-urban areas. This study used validated tools to assess women's knowledge about cervical cancer and screening, which increases the validity of this study's findings, with a representation of the rural, urban, and semi-urban comparability of this public health issue, hence this study offers valuable literature to inform design of targeted interventions, specifically for most at need geographical areas and populations. This study's findings are comparable with findings from a study conducted in Rwanda which revealed cervical cancer screening uptake was higher among women in urban districts compared to those living in the rural districts (9.3 vs 5.3) [5].

Our study findings are also supported by findings from a study conducted in United states which found that cervical cancer screening was lower in rural community health centers (CHCs) than urban CHCs (38.2% vs 43.0%) respectively during 2014–2019, and this difference increased during the pandemic (43.5% vs 49.0%), however, this study did not provide a comparison with the semi-urban setting. This difference is explained by the low education level and income below the poverty level in the rural areas [40]. This finding is also comparable with findings from a study in United Kingdom which revealed that cervical cancer screening rates were significantly lower among rural vs urban residents 48.6% vs 64.0%, P < 001 [41]. This finding is also supported by findings from a systematic review in Tanzania which found that women in urban areas felt more at risk compared to women in the rural areas (71% vs 62%) [42].

Women's history of smoking was found to be associated with cervical cancer screening uptake, suggesting a potential correlation between health-conscious behaviors and preventive health measures. This finding was not comparable with finding from a study in Jamaica which found that women who smoked were less likely to screen for cervical cancer screening [36]. Similarly, living with HIV was associated with higher prevalence of cervical cancer screening uptake. This finding is supported with findings from study conducted in Ethiopia which revealed that women living with HIV were 2 times more likely to screen for cervical cancer compared to HIV negative women [38].

Findings from this study are also comparable with findings from a study in Kenya which revealed that women living with HIV were 3 times more likely to screen for cervical cancer compared to HIV negative women [28]. These findings are also comparable with findings from a systematic review and meta-analysis in SSA which revealed that being HIV positive was a significant predictor of utilization of cervical cancer screening services [43–46]. These findings suggest that women living with HIV perceive themselves as at risk of acquiring cervical cancer, and hence go for cervical cancer screening services. Women with only one sexual partner in the past 12 months were less likely to undergo cervical cancer screening compared to those with no partners. This finding may reflect a perception of lower risk among monogamous individuals or potential barriers related to healthcare-seeking behaviors in relation to sexual health practices.

High knowledge about cervical cancer screening was strongly associated with increased uptake of cervical cancer screening, emphasizing the critical role of health education and awareness campaigns in promoting preventive health behaviors. Our study findings are comparable with findings from a systematic review and meta-analysis conducted in SSA

which found that knowledge about cervical cancer increased women's cervical cancer screening uptake by nearly five times [47], and the results are comparable with findings from a study in Ethiopia [45] which found that women who were knowledgeable were 4 times more likely to screen for cervical cancer [48].

In this study, women's knowledge regarding cervical cancer and screening may have significantly improved uptake of cervical cancer screening services in this population, and the high knowledge observed among women in this study suggests that women may be in position to recognize cervical cancer basing on its signs and symptoms and seek medical attention [24]. Also, if women are aware of the causes and risk factors of cervical cancer, and perceive themselves to be at risk, they are more likely to take up measures to prevent the acquisition of human papilloma virus hence avoid developing the disease.

Women who sought permission from their partners for healthcare decisions were significantly more likely to undergo screening and in the bivariate analysis, we found that women who reported joint decisions with their partners had a high uptake of cervical cancer screening in this population, highlighting the influence of social dynamics and support structures in healthcare utilization patterns, particularly partner support. This study also found that only 5% of the women reported receiving cervical cancer information from community opinion leaders, which highlights a gap in involvement of community opinion leaders in the effort to improve the country's cervical cancer screening rates.

## Conclusion

Awareness about cervical cancer and its screening is notably high among women in these communities in Wakiso district, Uganda. Majority of women demonstrated high knowledge about signs and symptoms of cervical cancer, and its risk factors. This suggests that cervical cancer information provided by healthcare workers when women come to seek other healthcare services (as observed in this study), and public health campaigns may have been effective in disseminating information about cervical cancer screening. This study found that healthcare workers were the primary source of information for cervical cancer screening which is critical for accurate information delivery and improving cervical cancer screening practices. However, this study also found that opinion leaders (religious leaders and teachers), mass media such as televisions and radios and the public were less involved in cervical cancer interventions, which leaves a blind spot in the on-going efforts to improve cervical cancer screening uptake in this setting. This study provides new knowledge in the cervical cancer landscape in Uganda, and informs an urgent need for community-engaged behavioral change interventions to modify individual behaviors, cultural and societal related barriers towards cervical cancer screening in Uganda. The magnitude of cervical cancer screening uptake in this study was lower than the 70% recommended coverage of the target group by the WHO and Uganda national guideline [2,49]. Older women, partner involvement, living with HIV, high knowledge about cervical cancer screening, having history of smoking and living in the urban areas were important factors associated with high uptake of cervical cancer screening in Wakiso district, Uganda [50–53].

### Limitations

While this study provides valuable insights in Uganda's cervical cancer landscape, several limitations should be considered. The cross-sectional nature limits causal inference, and potential biases such as recall and social desirability bias may have affected self-reported data on cervical cancer screening. Given the quantitative design of this study, the study did not provide an in-depth understanding of why the Knowledge, attitude, and practice of the participants are as they are. Future studies could go beyond this descriptive study to qualitatively explore the knowledge, attitude, barriers, and practice of women in relation to cervical cancer screening in Uganda.

## Supporting information

**S1 Dataset. Contains raw, anonymized data used for analysis in the study, including participant demographics, baseline scores, and outcomes.**
(CSV)

**S1 File. The full questionnaire used to assess perceptions of inclusivity in global health research among study participants has been also attached.**
(DOCX)

## Acknowledgments

The authors would like to thank the study participants for their time. We would also like to thank AMBSO's research team for their tireless efforts in the field. Sincere appreciations go to Wakiso District Local government, the NIHR, and RSTMH Early Careers Research Program for supporting the conduct of this research study.

## Author contributions

**Conceptualization:** Robert M. Bulamba.

**Data curation:** Robert M. Bulamba, Amanda Pearl Miller, Juliana Namutundu, Chris Balwanaki, Gertrude Nakigozi.

**Formal analysis:** Robert M. Bulamba, Alex Daama.

**Funding acquisition:** Robert M. Bulamba.

**Investigation:** Robert M. Bulamba, Fred Nalugoda, Godfrey Kigozi, Grace Kigozi Nalwoga, Stephen Watya, Anna Mia Ekström, Gertrude Nakigozi.

**Methodology:** Robert M. Bulamba, Fred Nalugoda, Alex Daama, Amanda Pearl Miller, William Byansi, Godfrey Kigozi, Grace Kigozi Nalwoga, Chris Balwanaki, Stephen Watya, Rawlance Ndejjo, Gertrude Nakigozi.

**Project administration:** Robert M. Bulamba, Emmanuel Kyasanku, Fred Nalugoda, James Nkale, Grace Kigozi Nalwoga, Stephen Watya, Stephen Mugamba, Gertrude Nakigozi.

**Supervision:** Robert M. Bulamba, Fred Nalugoda, Amanda Pearl Miller, William Byansi, Juliana Namutundu, Godfrey Kigozi, Grace Kigozi Nalwoga, Chris Balwanaki, Stephen Watya, Rawlance Ndejjo, Gertrude Nakigozi.

**Validation:** Juliana Namutundu, Rawlance Ndejjo, Gertrude Nakigozi.

**Visualization:** Robert M. Bulamba, James Nkale, Amanda Pearl Miller, William Byansi, Juliana Namutundu, Chris Balwanaki, Anna Mia Ekström, Rawlance Ndejjo, Gertrude Nakigozi.

**Writing – original draft:** Robert M. Bulamba.

**Writing – review & editing:** Robert M. Bulamba, Emmanuel Kyasanku, Fred Nalugoda, Alex Daama, James Nkale, Amanda Pearl Miller, William Byansi, Juliana Namutundu, Godfrey Kigozi, Grace Kigozi Nalwoga, Chris Balwanaki, Anna Mia Ekström, Stephen Mugamba, Rawlance Ndejjo, Gertrude Nakigozi.

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
