## [Decision Letter · Decision Letter 0]

17 Apr 2025

PONE-D-24-60418Assessing Knowledge, Uptake and Factors associated with cervical cancer screening among women in selected communities of Wakiso District in Uganda: A population-based studyPLOS ONE

Dear Dr. Bulamba,

Thank you for submitting your manuscript to PLOS ONE. After careful consideration, we feel that it has merit but does not fully meet PLOS ONE’s publication criteria as it currently stands. Therefore, we invite you to submit a revised version of the manuscript that addresses the points raised during the review process.

 Please submit your revised manuscript by Jun 01 2025 11:59PM. If you will need more time than this to complete your revisions, please reply to this message or contact the journal office at plosone@plos.org . Please include the following items when submitting your revised manuscript:

We look forward to receiving your revised manuscript.

Kind regards,

Zubing Mei, MD,PH.D

Academic Editor

PLOS ONE

Journal Requirements:

“This work was supported by the National Institute of Health and Care Research (NIHR), through the Royal Society of Tropical Medicine and Hygiene (RSTMH) Early Careers Research Program, grant number: NIHR170.”

4. Please note that your Data Availability Statement is currently missing the repository name and/or the DOI/accession number of each dataset OR a direct link to access each database. If your manuscript is accepted for publication, you will be asked to provide these details on a very short timeline. We therefore suggest that you provide this information now, though we will not hold up the peer review process if you are unable.

Additional Editor Comments:

The manuscript addresses an important public health issue—cervical cancer screening uptake in Uganda, a region with high cervical cancer incidence and mortality. The study’s focus on knowledge, uptake, and associated factors in diverse (urban, rural, and semi-urban) communities in Wakiso District is timely and relevant, given the low screening rates in low- and middle-income countries (LMICs). The use of a population-based approach and validated tools like the AWACAN instrument strengthens the study’s design. However, there are several areas where the manuscript could be improved to enhance its scientific rigor, clarity, and contribution to the field. Here are some major concerns to be addressed:1. The manuscript states a calculated sample size of 806 but does not provide the formula, assumptions (e.g., expected prevalence, precision, design effect), or justification for this number. This is critical for assessing the study’s power.2.The study achieved a sample of 783 women, but it is unclear whether this reflects a shortfall from the target (806) due to non-response or other factors. Non-response bias could affect generalizability.3. The manuscript mentions face-to-face interviews with a paper-based questionnaire, but it does not address interviewer training, potential interviewer bias, or measures to ensure data quality (e.g., double-entry, pilot testing).4. While the AWACAN tool is validated, the manuscript does not specify how it was adapted for this study (e.g., cultural or linguistic modifications) or whether its validity was re-assessed in this context.5. The criteria for including variables in the multivariable model are not specified. Were all variables from the bivariate analysis included, or was a p-value threshold (e.g., p<0.2) used? This lack of clarity raises concerns about model overfitting or omission of important confounders.

6.  The manuscript does not discuss potential confounders beyond those included in the model (e.g., access to healthcare facilities, income, or cultural beliefs). For example, the association between urban residence and screening uptake may be confounded by healthcare access, which is not adequately addressed.

7. The manuscript does not report checks for multicollinearity among predictors (e.g., between knowledge of cervical cancer and screening knowledge), which could affect the stability of regression estimates.

8. The use of the median to dichotomize knowledge scores (low vs. high) is arbitrary and may lose information. Continuous scores or alternative categorizations (e.g., tertiles) could provide more granularity.

9.The manuscript reports p-values of 0.000, which is imprecise. Exact p-values (e.g., p<0.001) or confidence intervals should be reported to avoid overstatement of significance.

10. The manuscript occasionally uses causal language (e.g., “associated with higher uptake”), which is inappropriate for a cross-sectional study. Associations do not imply causation, and this should be explicitly acknowledged.

11.  The discussion sometimes overstates findings. For example, the claim that partner involvement “significantly improves” screening uptake implies a causal effect that cannot be confirmed. Similarly, the association with smoking is interpreted as reflecting “health-conscious behaviors”, but alternative explanations (e.g., smoking as a proxy for other risk behaviors) are not explored.

12.  The discussion does not sufficiently compare findings with global literature, particularly from other LMICs or high-income settings. For example, how do the urban-rural differences align with studies in other SSA countries?

13. The limitations section is underdeveloped. Key limitations, such as potential recall bias in self-reported screening, social desirability bias, and the cross-sectional design’s inability to assess temporality, are not adequately discussed.

Reviewers' comments:

Reviewer's Responses to Questions

**Comments to the Author**

1. Is the manuscript technically sound, and do the data support the conclusions?

Reviewer #1: Yes

Reviewer #2: Yes

2. Has the statistical analysis been performed appropriately and rigorously? 

Reviewer #1: Yes

Reviewer #2: Yes

3. Have the authors made all data underlying the findings in their manuscript fully available?

Reviewer #1: Yes

Reviewer #2: Yes

4. Is the manuscript presented in an intelligible fashion and written in standard English?

Reviewer #1: Yes

Reviewer #2: Yes

5. Review Comments to the Author

Reviewer #1: 1. Does the cervical screening age in Uganda start from age 25? trying to understand why the participant minimum age was 25, considering that there is evidence of early cervical cancer chances in young women especially with HIV infection. Do clarify in the methodology why the age stipulated.

2. What new cervical screening prevalence was the study trying to ascertain considering that there was already studies done to ascertain the screening prevalence around the different districts in the country? Do elaborate in the conclusion if this is new knowledge.

3. What new information has the study provided for the district where the study was conducted? if there is, it should be elaborated in the conclusion or discussion, otherwise it seems like a study done for convenience with no new information that will benefit the health district.

Reviewer #2: Kindly find the following comments to be addressed

Ethical reference given from ethical board for PLOS ONE its record

Pretest

Software Analysis like Epidata or SPSS Epinfo

Crude and Adjust Odds Ratio before logistic regression

6. PLOS authors have the option to publish the peer review history of their article (what does this mean? ). If published, this will include your full peer review and any attached files.

**Do you want your identity to be public for this peer review?** For information about this choice, including consent withdrawal, please see our Privacy Policy .

Reviewer #1: **Yes: ** Mantwa Chisale Mabotja

Reviewer #2: **Yes: ** Abreha Tsegay Gebreselassie

---

## [Author Response · Author response to Decision Letter 1]

19 May 2025

Editor

No. Comment Response

Introduction

1 The manuscript states a calculated sample size of 806 but does not provide the formula, assumptions (e.g., expected prevalence, precision, design effect), or justification for this number. This is critical for assessing the study’s power. Thank you. This was an oversight, however, it has been addressed, and the sample size estimation description has been added to the methods section as below;

The study sample size was determined using the Kish Leslie’s formula (1965).

Sample Size (n) = P (1-P) Z2

________

d2

Samples Size (n) = 0.5* (1-0.5)* (1.96)2

(0.05)2

Where P= estimated proportion of cervical cancer screening among women (50%), as this provided the maximum possible sample size for more precise estimates of the study findings, Z= statistic corresponding to 95% Confidence level=1.96 and d=margin of error (5%). Therefore, the sample, n=384 participants. To adjust for sampling design and improve the precision of our estimates, we calculated the design effect as follows; Design effect (Deff) = 1+ δ(n-1), where, δ=intercluster correlation (ICC) and n=number of clusters. We used δ=0.5 recommended in previous work by (Liljequist, Elfving and Roaldsen, 2019) as a moderately good ICC for estimating representative sample sizes in one time surveys. Deff = 1+ 0.5(3-1) = 2. Therefore, the Sample Size = 384* Deff (2) = 768. Estimated non response rate was 5%, making the total study sample size of 806 women. Overall, our calculated sample was 806 inclusive of the 5% (38) non-response rate.

2 The study achieved a sample of 783 women, but it is unclear whether this reflects a shortfall from the target (806) due to non-response or other factors. Non-response bias could affect generalizability. Thank you for this comment. This has been addressed with clarity. I have added a description text below; Overall, our calculated sample was 806 inclusive of the 5% (38) non-response rate. Although our sample size calculation yielded a total target sample of 768 participants, this sample was exceeded by 15 participants during study enrolment, hence obtaining a sample of 783 participants. For this analysis, all the 783 enrolled participants were included.

3 The manuscript mentions face-to-face interviews with a paper-based questionnaire, but it does not address interviewer training, potential interviewer bias, or measures to ensure data quality (e.g., double-entry, pilot testing). Thank you so much for this comment, we have addressed this comment by adding the paragraph as below;

Data was collected by 15 research assistants (10-females and 5 males). The research assistants (RAs) were trained for 5 days (1st - 5th April 2024) on principles of quantitative and survey research including data collection, consenting process, the research study objectives, and procedures for; sampling and assessing eligibility for study participation. For validation purposes and checking for adaptability and understanding of the questions, consistence and flow of the questions, the data collection tools (questionnaires) were pre-tested on 10 women participants who were not living within the communities where the study was conducted. The 10 participants were as well reimbursed for their time during the pilot testing of the tool. After the pre-testing any required adjustments were made and then the research team was re-trained on the adjustments in the tools.

4 While the AWACAN tool is validated, the manuscript does not specify how it was adapted for this study (e.g., cultural or linguistic modifications) or whether its validity was re-assessed in this context. Thank you so much for this comment. The tool was adapted as is since it has been used elsewhere in similar populations in resource limited settings. Additionally, in line 200–204, we described our assessment for the reliability and internal consistency of the tool to determine its effectiveness in assessing our outcome. The lines read as below;

The test reliability and internal consistency for knowledge of cervical cancer risk factors was 0.89, and suggested a strong internal consistency of the measure’s ability to effectively assess women’s knowledge about cervical cancer risk factors, consistent with 0.60 obtained in prior work[19]. Similarly, the reliability and internal consistency of the tool’s measure for knowledge of signs and symptoms was assessed, and a Cronbach’s alpha of 0.86 was obtained, consistent with 0.80 obtained in previous validation study[19]

5 The criteria for including variables in the multivariable model are not specified. Were all variables from the bivariate analysis included, or was a p-value threshold (e.g., p<0.2) used? This lack of clarity raises concerns about model overfitting or omission of important confounders Thank you so much for this comment. This was a critical omission. This has been addressed by adding the following paragraph;

Since the prevalence of cervical cancer screening among women was 33.4%, we fitted a modified Poisson multivariable regression model with robust standard errors to determine the factors associated with uptake of cervical cancer screening among women. After the bivariate analysis, all factors with a P-value <=0.05 were included in the final regression model as potential factors associated with the outcome, adjusting for to limit bias from confounding. In the presentation of results, unadjusted and adjusted Prevalence Ratios (uPRs) and (aPRs) were interpreted and reported with accompanying 95% confidence intervals.

6 The manuscript does not discuss potential confounders beyond those included in the model (e.g., access to healthcare facilities, income, or cultural beliefs). For example, the association between urban residence and screening uptake may be confounded by healthcare access, which is not adequately addressed. Thank you so much for this comment. We have strengthened our discussion and added the following text in the discussion;

The high rates of cervical cancer screening observed particularly in urban areas may have been influenced by availability and quick access to health care services in the urban areas compared to rural/semi-urban areas.

7 The manuscript does not report checks for multicollinearity among predictors (e.g., between knowledge of cervical cancer and screening knowledge), which could affect the stability of regression estimates. Thank you so much for this comment. This was an oversight. This has been addressed and the following text has been added under data preparation and statistical analysis in the methods section;

Multicollinearity between predictors such as overall knowledge for cervical cancer and knowledge for cervical cancer screening (was 2.07) and between knowledge of cervical cancer signs and symptoms with risk factors (was 1.14) was checked using the variation inflation factor (VIF), which suggested absence of multicollinearity between these predictors in line with previous models(26)

8 The use of the median to dichotomize knowledge scores (low vs. high) is arbitrary and may lose information. Continuous scores or alternative categorizations (e.g., tertiles) could provide more granularity. Thank you so much for this comment. We used the median approach, guided by previous work by; (Ahmed et al., 2023), (Mukama et al., 2017) and (Moodley et al., 2019) an appropriate technique for assessing knowledge of participants.

9 The manuscript reports p-values of 0.000, which is imprecise. Exact p-values (e.g., p<0.001) or confidence intervals should be reported to avoid overstatement of significance. Thank you so much for this comment. This has been addressed in the table and whenever applicable.

10 The manuscript occasionally uses causal language (e.g., “associated with higher uptake”), which is inappropriate for a cross-sectional study. Associations do not imply causation, and this should be explicitly acknowledged. Thank you so much for this comment. This has been addressed, and a study limitation section has been added where this has been fully acknowledged. See response in comment 11 below.

11 The limitations section is underdeveloped. Key limitations, such as potential recall bias in self-reported screening, social desirability bias, and the cross-sectional design’s inability to assess temporality, are not adequately discussed. Thank you so much for this comment. This has been addressed and a limitations section has been included in the manuscript. The text reads as follows;

While this study provides valuable insights, several limitations should be considered. The cross-sectional nature limits causal inference, and potential biases such as recall and social desirability bias may have affected self-reported data on cervical cancer screening. Given the quantitative design of this study, the study did not provide an in-depth understanding of why the Knowledge, attitude, and practice of the participants are as they are. Future studies could go beyond this descriptive study to qualitatively explore the knowledge, attitude, and practice of women in relation to cervical cancer screening.

Reviewer One

1 Does the cervical screening age in Uganda start from age of 25? trying to understand why the participant's minimum age was 25, considering that there is evidence of early cervical cancer chances in young women, especially with HIV infection. Do clarify in the methodology why the age stipulated. Thank you so much for this comment. Although presently the risk may be epidemiologically shifting to even the lower age groups, the current Uganda national guidelines prioritize prevention efforts among women starting from 25 years. This guided our study's target population. This has been clarified in the methods section as below under lines 104-105. We have added the following text;

Our study target population was guided by the present national guidelines which target screening women starting from 25 years.

2 What new information has the study provided for the district where the study was conducted? if there is, it should be elaborated in the conclusion or discussion, otherwise it seems like a study done for convenience with no new information that will benefit the health district. Thank you so much for this comment. Lines 78-84 strengthens the need for the conduct of this study. The text reads as follows;

This study also found that opinion leaders (religious leaders and teachers), mass media such as televisions and radios and the public were less involved in cervical cancer interventions, which leaves a blind spot in the on-going efforts to improve cervical cancer screening uptake in this setting. This study provides new knowledge in the cervical cancer landscape in Uganda, and informs a urgent need for community-engaged behavioral change interventions to modify individual behaviours, cultural and societal related barriers towards cervical cancer screening in Uganda.

Additionally, there exists limited population-level cervical cancer data in Uganda, particularly in Wakiso district, which limits the design of high-impact community-based cervical cancer screening interventions to reduce morbidity and mortality in the district. The Uganda MOH recommends cervical cancer screening for women aged 25-49 years, however, the current WHO guidelines highlights a gap in data regarding cervical cancer screening among women aged 50-65 years, hence the inclusion of this population strengthens the relevance of our study by offering insightful literature in the Ugandan context.

Reviewer 2

2 Pretest This has been addressed. We have added the text. See response to comment 3 above.

3 Software Analysis like Epidata or SPSS Epinfo Thank you so much for this comment. In line 207, we indicate the software used to perform our analysis.

4 Crude and Adjust Odds Ratio before logistic regression Thank you so much for this comment. Given the observed prevalence of our outcome (33.4%), a Poisson regression model was an appropriate statistical test in comparison to the Logistic regression technique.

---

## [Decision Letter · Decision Letter 1]

12 Jun 2025

Assessing Knowledge, Uptake and Factors associated with cervical cancer screening among women in selected communities of Wakiso District in Uganda: A population-based study

PONE-D-24-60418R1

Dear Dr. Bulamba,

We’re pleased to inform you that your manuscript has been judged scientifically suitable for publication and will be formally accepted for publication once it meets all outstanding technical requirements.

Kind regards,

Zubing Mei, MD,PH.D

Academic Editor

PLOS ONE

Additional Editor Comments (optional):

Reviewers' comments:

Reviewer's Responses to Questions

**Comments to the Author**

1. If the authors have adequately addressed your comments raised in a previous round of review and you feel that this manuscript is now acceptable for publication, you may indicate that here to bypass the “Comments to the Author” section, enter your conflict of interest statement in the “Confidential to Editor” section, and submit your "Accept" recommendation.

Reviewer #1: All comments have been addressed

Reviewer #2: All comments have been addressed

2. Is the manuscript technically sound, and do the data support the conclusions?

Reviewer #1: Yes

Reviewer #2: Yes

3. Has the statistical analysis been performed appropriately and rigorously? 

Reviewer #1: I Don't Know

Reviewer #2: Yes

4. Have the authors made all data underlying the findings in their manuscript fully available?

Reviewer #1: Yes

Reviewer #2: Yes

5. Is the manuscript presented in an intelligible fashion and written in standard English?

Reviewer #1: Yes

Reviewer #2: Yes

6. Review Comments to the Author

Reviewer #1: The authors has addressed my comments raised previously, if the statistical comments made by the other reviewer have been addressed to the reviewers satisfaction then I accept to have the manuscript published.

Reviewer #2: I can see the detailed comments provided by reviewer to be addressed by authors are well addressed each comments

7. PLOS authors have the option to publish the peer review history of their article (what does this mean? ). If published, this will include your full peer review and any attached files.

**Do you want your identity to be public for this peer review?** For information about this choice, including consent withdrawal, please see our Privacy Policy .

Reviewer #1: **Yes: ** Mantwa Chisale Mabotja

Reviewer #2: **Yes: ** Abreha Tsegay Gebreselassie

---

## [Editor Report · Acceptance letter]

PONE-D-24-60418R1

PLOS ONE

Dear Dr. Bulamba,

I'm pleased to inform you that your manuscript has been deemed suitable for publication in PLOS ONE. Congratulations! Your manuscript is now being handed over to our production team.

Kind regards,

on behalf of

Dr. Zubing Mei

Academic Editor

PLOS ONE